# Adverse Effects of Using Metallurgical Slags as Supplementary Cementitious Materials and Aggregate: A Review

**DOI:** 10.3390/ma15113803

**Published:** 2022-05-26

**Authors:** Qiang Zhao, Lang Pang, Dengquan Wang

**Affiliations:** 1Beijing Urban Construction Group, Beijing 101499, China; bjcjadzq@mail.bucg.com; 2School of Civil Engineering, Beijing Jiaotong University, Beijing 100091, China; 20121088@bjtu.edu.cn; 3Department of Civil Engineering, Tsinghua University, Beijing 100084, China

**Keywords:** metallurgical slags, supplementary cementitious material, aggregate, microstructure, mechanical properties, safety

## Abstract

This paper discusses a sustainable way to prepare construction materials from metallurgical slags. Steel slag, copper slag, lead-zinc slag, and electric furnace ferronickel slag are the most common metallurgical slags that could be used as supplementary cementitious materials (SCMs) and aggregates. However, they have some adverse effects that could significantly limit their applications when used in cement-based materials. The setting time is significantly delayed when steel slag is utilized as an SCM. With the addition of 30% steel slag, the initial setting time and final setting time are delayed by approximately 60% and 40%, respectively. Because the specific gravity of metallurgical slags is 10–40% higher than that of natural aggregates, metallurgical slags tend to promote segregation when utilized as aggregates. Furthermore, some metallurgical slags deteriorate the microstructure of hardened pastes, resulting in higher porosity, lower mechanical properties, and decreased durability. In terms of safety, there are issues with the soundness of steel slag, the alkali-silica reaction involving cement and electric furnace ferronickel slag, and the environmental safety concerns, due to the leaching of heavy metals from copper slag and lead-zinc slag.

## 1. Introduction

Cement is the most widely used construction material, with over 3 billion t of production worldwide each year [1]. The production of cement is an energy-intensive process and produces large amounts of CO_2_ emission, which causes serious environmental problems of global warming and climate change [2,3]. Therefore, cement-based products must be made more environmentally friendly [4]. Some metallurgical slags (e.g., steel slag, copper slag, lead-zinc slag, and electric furnace ferronickel slag) have the potential to be used in cement-based materials, due to their similar composition and morphology to cement or sand. Because of the huge consumption of construction materials, the partial substitution of cement or sand with metallurgical slags can significantly reduce CO_2_ emissions and save natural resources, while also avoiding the discharge and pollution of waste slags.

The basic oxygen furnace steel slag (abbreviated as SS) is a solid by-product released during steelmaking that accounts for 15–20% of crude steel production [5,6,7,8]. The following content is focused on the basic oxygen furnace steel slag. Approximately 200 million t of SS are produced worldwide each year, and China accounts for half of them [9,10]. As shown in Table 1, the chemical components in SS mainly contain CaO, Fe_2_O_3_, SiO_2_, Al_2_O_3_, MnO, and MgO [11]. The main mineral phases are C_3_S, C_2_S, C_2_F, and RO phase (solid solution consisting of divalent metal oxides, such as CaO, FeO, MgO, and MnO) [12,13,14]. SS has a long history of being used as a construction material, due to its cementitious properties [15].

Copper slag (CS) is a by-product of copper manufacturing, with 2.2 t of CS released from every ton of copper produced. In 2020, more than 20 million t of CS were generated worldwide [16,17]. In China, the accumulated stockpile of CS has reached 130 million t, with 80% of the CS remaining unutilized, owing to a lack of relevant technologies [18]. The main chemical components in CS include Fe_2_O_3_, SiO_2_, Al_2_O_3_, CaO, ZnO, MgO, SO_3_, CuO [19]. The main mineral phases are Fe_2_SiO_4_, Fe_3_O_4_, FeO, Fe_2_O_3_, ZnO, FeAl_2_O_4_, 2MgO∙SiO_2_ and MgAl_2_O_4_ [20]. The concentration of glass in different kinds of CS varies substantially due to the diverse cooling processes, resulting in different ways of utilization. Currently, CS can be used as fine aggregate [21,22,23,24,25,26] or supplementary cementitious material [27,28,29].

Lead-zinc slag (LZS) is the waste that is discharged during the extraction of lead (Pb) and zinc (Zn). For every ton of lead and zinc produced, 710 kg and 960 kg of LZS are generated, respectively. According to the statistics, the annual emissions of LZS have exceeded 5.5 million t worldwide [30,31], which requires a significant amount of land to stockpile [32]. The chemical components of LZS generally consist of Fe_2_O_3_, CaO, SiO_2_, Al_2_O_3_, MgO, PbO, and ZnO. The mineral phase may include Fe_2_SiO_4_, CaFeSiO_4_, Zn_2_SiO_4_, (Zn, Mg, Fe)(Fe, Al, Cr)_2_O_4_, Fe_3_O_4_, Fe_2_O_3_, PbO, PbS, ZnS and Pb [30]. Furthermore, LZS contains some glass as a result of the cooling process of water quenching from the molten state. LZS can be used as fine aggregate in road foundations [33].

Ferronickel slag is a granulated solid waste formed by cooling the molten slag that is discharged after the extraction of nickel and iron during the smelting process of ferronickel alloys [34]. The production of 1 ton of ferronickel alloy will generate 14 t of slag [35], and the annual emissions of ferronickel slag have exceeded 30 million t in China, accounting for 20% of the global production [36]. Ferronickel slag can be classified into two types based on its smelting technique and equipment, namely electric furnace ferronickel slag (EFS) and blast furnace ferronickel slag (BFS). The chemical composition of EFS is mostly SiO_2_, Fe_2_O_3_, and MgO, with a little Al_2_O_3_ and CaO [37]. The mineral composition is primarily Mg_2_SiO_4_, with some amorphous glass [38]. BFS is mostly composed of SiO_2_, Al_2_O_3_, and CaO, with trace amounts of MgO and Fe_2_O_3_; the mineral composition is primarily composed of MgAl_2_O_4_, MgSiO_3_, Ca_2_SiO_4_, CaCO_3_, and amorphous glass [39,40,41,42,43]. As the electric furnace is the most commonly used technique of manufacturing, this work focuses only on electric furnace ferronickel slag. EFS can be used as an SCM and fine aggregate in construction materials [44].

Steel slag, copper slag, lead-zinc slag, and electric furnace ferronickel slag are the most common metallurgical slags. There are numerous precedents of these slags being applied to cement-based materials due to their physical and chemical qualities [44,45,46,47,48,49,50]. However, the use of these metallurgical slags in cement-based materials has several adverse effects that significantly restrict their utilization. This study covers the previous investigations of the unfavorable impacts of these metallurgical slags in cement-based materials, such as fresh properties, hydration process, microstructure, mechanical properties, durability, and safety. This work can help to advance understanding in this field, as well as to encourage the exploration of metallurgical slags and the sustainable development of construction materials.

**Table 1 materials-15-03803-t001:** Chemical components of these metallurgical slags (wt/%).

Compound	SS		CS		LZS		EFS	
	Average	Range	Average	Range	Average	Range	Average	Range
SiO_2_	16.40	11.04–23.30	29.09	22.63–33.62	25.30	18.89–30.76	41.22	32.74–53.29
Al_2_O_3_	3.56	1.61–6.10	4.76	2.79–8.25	5.08	3.44–7.28	8.23	2.67–16.52
Fe_2_O_3_	24.52	16.85–31.60	54.26	49.88–61.70	34.64	28.10–46.81	26.04	9.57–43.83
CaO	40.54	30.80–45.26	3.42	1.87–6.06	15.12	10.53–23.11	4.20	0.42–11.49
MgO	7.55	5.98–12.00	1.49	0.99–1.81	2.80	1.28–5.44	16.07	2.76–32.88
Na_2_O	0.33	0.20–0.45	0.41	0.14–0.96	1.44	0.27–4.12	0.33	0.09–0.80
K_2_O	0.07	0.07	1.08	0.61–2.28	0.59	0.26–0.96	0.28	0.18–0.37
SO_3_	0.15	0.11–0.19	1.67	1.12–1.99	3.97	2.41–6.64	0.58	0.52–0.64
TiO_2_	0.89	0.50–1.57	0.00	0.00	0.56	0.19–0.90	0.18	0.12–0.24
P_2_O_5_	1.49	0.01–3.24	0.05	0.05–0.05	0.22	0.15–0.36	1.86	1.86–1.86
MnO	2.26	1.50–3.04	0.06	0.06–0.06	1.83	0.66–2.97	0.58	0.44–0.81
CuO	-	-	1.60	0.57–2.63	0.53	0.42–0.62	-	-
Cr_2_O_3_	0.03	0.15	-	-	0.14	0.11–0.19	1.97	0.70–3.07
PbO	-	-	-	-	1.86	0.03–4.06	-	-
ZnO	-	-	2.31	2.31–2.31	8.25	5.01–13.95	-	-
LOI	1.25	0.64–1.86	6.09	6.09–6.09	4.52	0.46–7.48	3.44	3.44

Note: Date from: (1) Steel slag: Liao et al. [13]. Mo et al. [51]. Shi et al. [52]. Wang et al. [53]. Muhmood et al. [54]. (2) Copper slag: Zhang et al. [55]. Ahirwar et al. [56]. Siddique et al. [57]. Lan et al. [58]. Gupta et al. [59]. Al-Jabri et al. [28]. (3) Lead-zinc slag: Xia et al. [60]. Mao et al. and Li et al. [61,62]. Lima et al. [63]. Saikia et al. [64]. Weeks et al. [65]. Penpolcharoen, [66]. (4) Electric furnace ferronickel slag: Luo et al. [67]. Cao et al. [68]. Rahman and Saha et al. [69,70]. Lemonis and Katsiotis et al. [71,72]. Maragkos et al. [73]. Komnitsas et al. [74].

## 2. Fresh Properties

### 2.1. Metallurgical Slags as SCMs

Some metallurgical slags could delay the early setting of composite paste when used as SCMs. In the study of Zhuang et al. [75], 3 types of SS powder were used to replace 30% of cement. The initial setting time of the paste was prolonged from 174 min of the control group to 495, 476, and 423 min, respectively, with an average increase of 2.67 times. For CS, Zain et al. [76] found that when CS was used as an SCM, it also increased the setting time. However, some researchers have reached the opposite conclusion. Gopalakrishnan et al. [77] found that the addition of CS shortened the setting time, which can be attributed to the influences of different production procedures, fineness, water demand, and other factors of CS. In terms of EFS, Kim et al. [78] measured the setting time of an EFS-containing mortar with penetration resistance. The results showed that the initial and final setting times were 100 and 140 min longer than those of the control group at 30% of EFS incorporation, both exceeding 26%.

Figure 1 shows the influences of these metallurgical slags on the setting time of composite paste when they are used as SCMs. It can be found that the delaying effect of SS on setting time is the most obvious (especially for the initial setting time). With SS blending at 30%, the initial setting time is delayed by about 60%, and the final setting time is delayed by about 40%. CS has the least impact on setting time. In addition, the impact of EFS on the final setting time is more significant than that on the initial setting time. With EFS doping at 30%, the final setting time is prolonged by about 10%.

### 2.2. Metallurgical Slags as Aggregates

The specific gravity of metallurgical slags is typically 10–40% greater than that of natural aggregates (Table 2). These slags may cause bleeding or segregation of the paste when they are used as aggregates. In addition, some metallurgical slags can reduce the fluidity or prolong the setting time of paste.

Wang proposed that the replacement rate of CS to sand should be less than 40% to avoid bleeding [85]. This may be attributed to the wrong mixing water content or the lower water absorption of CS (it increases the free water content in the paste) [27]. It has been found that EFS causes serious bleeding and segregation when it is used as an aggregate [86]. The bleeding is caused by the upward migration of water, and the early hydration products will block the migration channel and, therefore, halt the bleeding. The early hydration products are greatly reduced and the setting time is prolonged due to the addition of metallurgical slags. The plastic stage of the paste becomes longer, so that the duration of bleeding will also be extended [87].

Mosavinezhad et al. [88] replaced 30% of the natural sand with lead slag and zinc slag, respectively, as fine aggregate for concrete. The results showed that the slump of fresh concrete was reduced from 135 mm to 75 mm and 20 mm, respectively, reduced by 44% and 85%, indicating that the fluidity of concrete significantly decreased. This is because the lead slag and zinc slag have a lower fineness modulus with more angular shapes. Saikia et al. [89] discovered that substituting natural fine aggregate with LZS delayed the setting time of composite paste (Figure 2), and the setting time grew as the LZS concentration increased. With lead and zinc slag doping of 35%, the initial setting time and final setting time were extended from 323 min and 405 min to 787 min and 992 min, respectively, with an average extension of 2.4 times.

**Table 2 materials-15-03803-t002:** Specific gravity and water absorption of the four metallurgical slags.

Slag	Ref	Specific Gravity (g/cm^3^)		Water Absorption (%)
		Slag	Natural Aggregate	
SS	Lim et al. [90]	3.56	2.65	1.5
	Palankar et al. [91]	3.35	2.69	2
	Qasrawi, [92]	3.19	2.57	0.8
CS	Sim et al. [93]	3.53	2.51	0.16
	Sharma et al. [23]	3.51	2.6	0.36
	Patil et al. [94]	3.3	2.65	0.65
LZS	Penpolcharoen [66]	3.62	2.71	-
	Buzatu et al. [32]	3.79	2.65	1.29
	Saikia et al. [89]	3.76	2.62	1.5
EFS	Nguyen et al. [37]	2.88	2.65	0.8
	Sun et al. [44]	2.99	2.65	0.94
	Saha et al. [69]	2.78	2.16	0.42

## 3. Hydration Process

The impacts of metallurgical slags on the hydration process of cement are most visible when used as SCMs, and less noticeable as aggregates. This is mainly because of the inhibition effect of metallurgical slags on the hydration of cement, which can be further elaborated as the following three reasons: (1) metallurgical slags inhibit the generation of hydration products of cement; (2) some compounds in metallurgical slags can encapsulate the clinker particles and hinder the hydration; (3) metallurgical slags alter the morphology and composition of hydration products.

Zhuang et al. [75] found that SS can significantly inhibit the early-age hydration of cement. The calorimetry curves are shown in Figure 3. It can be observed that the addition of SS significantly reduces the cumulative heat release and delays the main exothermic peak. This is because the SS inhibits the nucleation and growth of ettringite, C-S-H, and other hydration products (Figure 4). As a result, the crystallization of these hydration products and the formation of the solid network that provides the initial strength are delayed. This study also found that the low alkali content of SS leads to a low pH value of composite paste. On the one hand, the low pH causes the supersaturation of Ca(OH)_2_, which is lower than the critical value and limits its precipitation. On the other hand, it raises the concentration of Ca^2+^, shifting the dissolving equilibrium of formula (1) to the left, thus, lowering the consumption of gypsum. Therefore, SS can significantly inhibit the hydration of cement, which also explains the delaying effect of SS on the setting time in Section 2.1.
CaSO_4_(s) = Ca^2+^(aq) + SO_4_^2−^(aq)(1)

Some studies revealed that heavy metal elements, such as lead, zinc, and copper, could inhibit the hydration of cement [95,96,97]. This is due to the fact that these heavy metal compounds can form a crystalline film on the surface of clinker particles, wrapping the clinker particles and hindering hydration. For example, it was found that the Cu, Pb, and Zn compounds in CS cover the silicate phases of cement clinker, hence, inhibiting the hydration [76,98]. Saca et al. [99] replaced cement clinker with LZS and found that the hydration degree of the binder decreased when the admixture was increased from 30% to 50%. Saikia et al. [89] attributed this to the presence of Pb and Zn-containing phases. When these phases dissolved at a high pH value, Pb and Zn formed some insoluble compounds, as well as a thin film to seal the clinker particles, which eventually inhibits the hydration of cement.

The research results of Li et al. [100] showed that pure cement paste generated clustered hydration products (C-S-H and CH) at 28 days, while the paste containing EFS produced only CH with sizes less than 1 μm and many spherical C-S-H (Figure 5). This is due to the low CaO content and high Al_2_O_3_ content of EFS. In composite paste, EFS decreases the Ca/Si ratio of the C-S-H gel and causes it to contain the aluminum (Al) phase [100,101]. Therefore, EFS will change the morphology and composition of cement hydration products.

## 4. Microstructure, Mechanical Properties and Durability

The microstructure of hardened paste plays a decisive role in mechanical properties and durability. When metallurgical slags are added to cement-based materials, they usually deteriorate the microstructure of hardened paste and eventually reduces its strength and durability. When metallurgical slags are used as SCMs, the reduction in the cementitious composition and the inhibition effect on the hydration of cement will reduce the compactness of hardened paste. When metallurgical slags are used as aggregates, the uniformity and compactness of hardened paste are degraded due to the bleeding effect.

### 4.1. Steel Slag

As described in Section 2.1, when SS is used as an SCM, it retards the hydration of cement and reduces the formation of early-age hydration products. This adverse effect can delay the development of microstructures in early age [75]. As shown in Figure 6, the relative porosity of the paste is calculated after using SS as an SCM in the existing studies. It can be observed that the porosity gradually increases with the increase in the content of SS, and the porosity increases by nearly 20% when the content is at 30%. Figure 7 shows the 28 d relative compressive strength of the hardened paste after adding SS powder. With the increase in SS content, the relative strength gradually decreases, and when the content is 30%, the relative strength decreases by nearly 20%. According to the research of Wang et al. [102,103], the porosity increased from 9.7% to 11.3% at 90 days when using SS to replace 45% of cement. This reduces the strength of hardened paste (especially the early strength). The greater the water-to-binder ratio, the more obvious the reduction. The author described that this was because of the low cementitious properties of SS. It was also found that the impermeability of hardened paste decreased after adding SS. In addition, the carbonization depth increased from 4.5 mm to more than 12 mm when the water-to-binder ratio was 0.5. Liu et al. [104] found similar results.

When SS is used as an aggregate, it has been found that SS increases the porosity of the paste [105], while decreasing the strength of the interfacial transition zone, thus, reducing the mechanical properties of the bulk [106].

**Figure 6 materials-15-03803-f006:**
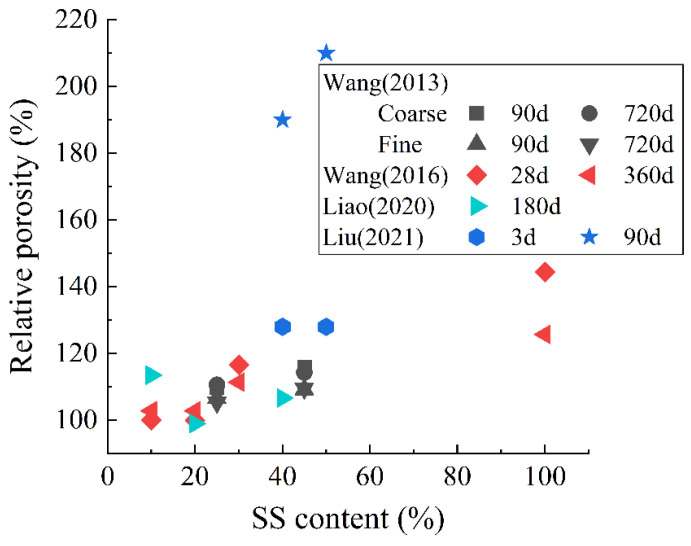
Relative porosity of hardened paste with SS as an SCM [12,13,102,104]. The relative porosity is the ratio of the porosity of composite paste containing metallurgical slag to that of the plain cement paste.

**Figure 7 materials-15-03803-f007:**
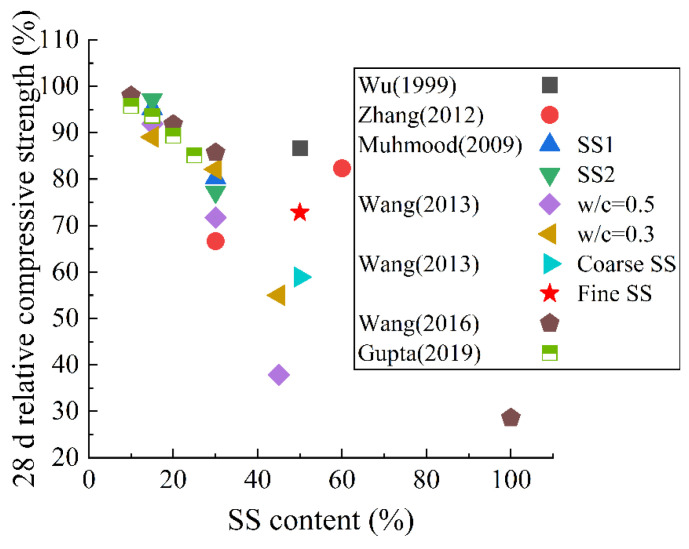
The 28 d relative compressive strength of hardened paste with SS as an SCM [12,54,80,82,102,103,107]. The relative compressive strength is the ratio of the compressive strength of composite paste containing metallurgical slag to that of the plain cement paste.

### 4.2. Copper Slag

When CS is used as an SCM, different researchers have proposed inconsistent results regarding the effect of CS on the pore structure of the hardened composite paste. The research of Zhang et al. [55] showed that adding CS can increase the porosity of the composite paste, while Gopalakrishnan et al. [77] found that the porosity decreased with the increase in the content of CS. Boakye et al. [108] revealed that when the content of CS was 2.5%, 5%, 10%, and 15%, the water absorption of hardened paste decreased by 4%, 7%, 9%, and 11%, respectively. Figure 8 shows the 28 d relative compressive strength of hardened paste after adding CS. It can be observed that the laws of most research results are similar. With the increase in CS content, the relative strength decreases linearly. When the content is 30%, the 28 d compressive strength decreases by about 20%. However, some researchers have achieved different outcomes. For example, the research of Zheng et al. [16] showed that the strength was improved when the content of CS was 10% or 20%. Various investigations discovered that the variation in the influence of CS on the porosity and compressive strength of hardened paste can be ascribed to the difference in manufacturing technique, chemical and mineral composition, and copper slag grinding procedure [109].

When CS is used as an aggregate, many studies have proposed a positive effect of CS on the concrete, while some other studies have found an adverse effect. This is related to the different nature of CS from different areas, as well as the different mix proportions in different studies. Although copper slag has numerous positive advantages when utilized as an aggregate, its potential negative impacts must not be overlooked. Maharishi et al. [113] found that the adverse effect of CS on concrete was particularly significant when the content of CS exceeded 40%. Patil et al. [94] discovered that adding CS caused greater porosity in the microstructure (Figure 9), owing to the poor connectivity of CS and cement. On the aspect of mechanical properties, Al-Jabri et al. [114] and Kubissa et al. [115,116] found that adding CS can effectively improve the mechanical strength of concrete. However, some studies found the opposite results. Sridharan et al. [117] found that the strength of hardened paste began to reduce when the content of CS exceeded 40% (especially the flexural strength). Similarly, the research of Wu et al. [118] and Sambangi et al. [24] also showed that the early strength reduced when the content of CS exceeded 40%. The 3 d compressive strength and flexural strength were reduced by about 30% when the content of CS was 100% [24]. Some other studies also found this phenomenon [119,120,121]. For durability, Kubissa et al. [115] found that CS reduced the sorptivity and improved the permeability resistance of concrete. In contrast, Patil et al. [94] found that the impermeability, chloride ion corrosion resistance, and sulfate attack resistance of hardened paste decreased after adding CS. When the substitution rate was set at 40%, the permeability increased by more than 50%. Kranti et al. [120] also found that the sulfate attack resistance of hardened paste was reduced by replacing sand with CS. Moreover, the strength of hardened paste decreased significantly after the sulfate attack [121]. This is because the porous concrete formed by adding CS is soaked in sulfate. Internal stress caused by sulfate crystal precipitation and the production of ettringite may degrade the microstructure and potentially induce cracking. Furthermore, the decomposition of Ca(OH)_2_, and the scaling and softening of gypsum can degrade or disintegrate the composite paste [121,122].

### 4.3. Lead-Zinc Slag

Saca et al. [99] found that the strength of hardened paste decreased after replacing cement clinker with LZS. When the content of LZS increased from 30% to 50%, the 28 d compressive strength decreased from 39.7 MPa to 25.8 MPa, reduced by 35%; the flexural strength reduced from 7 MPa to 5.6 MPa, reduced by 20%. This is because the paste forms a weaker microcrystalline aspect after mixing with LZS.

Figure 10 and Figure 11 calculate the relative porosity and 28 d relative compressive strength of the hardened paste containing LZS aggregate. It can be observed that the porosity tends to increase when LZS is used as an aggregate. According to the findings of Saikia et al. [89], the porosity of 28 d hardened paste rises from 17.32% to 21.03%, and the water absorption rises from 8.29% to 8.75% when the LZS admixture increases from 0 to 35%. Regarding the mechanical properties, Figure 11 shows that the results vary from different researchers. Saikia et al. [89] found that the addition of LZS reduced the compressive strength. Mosavinezhad et al. [88] used zinc slag to replace fine aggregate sand by 30%, and the results showed the compressive strength and flexural strength of hardened paste in 56 days decreased by about 88% and 70%, respectively. In contrast, Alwaeli [46] revealed that the addition of LZS improved the strength of the concrete. When the content of LZS was 25%, 50%, 75%, and 100%, the compressive strength increased by 5.90%, 13.23%, 16.40%, and 33.60%, respectively. This discrepancy is due to the changes in chemical composition, fineness, and cementitious activity of various LZS.

### 4.4. Electric Furnace Ferronickel Slag

EFS powder can be used as an SCM. Figure 12 shows the effect of EFS content on the 28 d compressive strength. Many researchers have discovered that as the content of EFS powder increases, the compressive strength of hardened paste decreases linearly. When the content is 30%, the 28 d compressive strength decreases by about 15%. The research of Li et al. [100] showed that the 28 d hardened paste containing 20% EFS had 2.39% more harmful pores (d > 50 nm) than those of the pure cement group. This is consistent with the result for compressive strength. There is a 22.5% reduction in 28 d compressive strength from 60 MPa to 46.5 MPa [100].

## 5. Safety

### 5.1. Soundness

Many studies have shown that the basic oxygen furnace steel slag has a high concentration of free CaO and MgO. When SS is utilized as an aggregate, the reaction of free CaO and MgO will cause bad soundness, which will seriously threaten the safety of constructions [91,123,124,125,126]. The results of Palankar et al. [91] showed that the pastes containing SS aggregate developed a soundness problem, which was due to the expansion of SS. As shown in Figure 13, steel slag aggregate appears to deteriorate due to internal pressure.

### 5.2. Alkali-Silica Reaction

EFS contains a large amount of amorphous silica, which is prone to causing alkali-silica reactions and volume expansion when used as an aggregate in cement-based materials [35]. As shown in Figure 14 and Figure 15, Choi et al. [127] discovered that when water-cooled EFS was used to substitute natural aggregates, the alkali-silica reaction resulted in significant volume expansion and micro fractures. The expansion ratio of 28 d hardened mortar prepared with water-cooled EFS was between 0.6% and 0.8%, while it was lower than 0.05% when prepared with air-cooled EFS. On the one hand, the rapid cooling of EFS by water causes the slag particles to exhibit more micro cracks, thus, accelerating the dissolution and reaction of silica-containing phases. On the other hand, the rapid cooling leads the silicon phase to become less crystalline and produces more amorphous silica [86,127].

### 5.3. Environmental Safety

The environmental problems of copper slag used in cement-based materials are mainly focused on the leaching of heavy metals (Pd, Cd, As, Cu, Zn, etc.) and the potential radiological hazards [128]. Although Kubissa et al. [129] have revealed that the radioactivity of concrete containing CS is low, the potential hazards cannot be ignored. Wang et al. [27] studied the performance of CS from 10 different production areas as SCMs. CS poses a great radiological risk (^226^Ra, ^232^Th, and ^40^K) when the content reaches 50% (Figure 16). In addition, there is a leaching risk of heavy metals, such as Cu, Pb, Zn, Ni, and As. As shown in Figure 17, when the CS content is 30%, the leaching amount of Cu and Pb elements exceeds the safety limit. At a replacement rate of 50%, Zn, Ni, and As also result in leaching hazards. Similarly, He et al. [112] found that CS posed a leaching risk of heavy metals (Cu, Pb, Mn, Zn, Ni, Cd, and Cr). The author also pointed out that the main crystalline phase in CS is fayalite. As shown in Figure 18, it dissolves quickly in acidic conditions but not so easily in alkaline conditions. This is owing to the presence of adsorption and precipitation in an alkaline environment, which will decrease the leaching of heavy metals. For example, the leaching amount of Pb in an acidic environment increases significantly because there is a different leaching control phase of Pb in different pH environments. When the pH is < 5, the main leaching control phase is anglesite (PbSO_4_), while it is laurionite (Pb(OH)Cl) in an alkaline environment. The leaching of Cu in an alkaline environment is mainly controlled by Cu(OH)_2_ and Cu_2_CO_3_(OH)_2_ with a low solubility, so the leaching amount of Cu in an alkaline environment is very low. In acidic conditions, those Cu that adsorbed by Al(OH)_3_ and Fe(OH)_3_ via surface complexation is easy to be eroded by H^+^, resulting in increased leaching. In addition, Mn, Zn, Ni, and Cd have similar variation rules, while Cr has a high leaching rate in an alkaline environment. This is because the main precipitation of Cr (III) in an alkaline environment is Cr(OH)_3_. It is an amphoteric hydroxide and could be dissolved in the form of a complex in a highly alkaline environment. Moreover, Cr (VI) usually does not produce precipitation in the leaching system [130,131,132,133].

When LZS is used in cement-based materials, there is a leaching risk of harmful elements, such as Zn, Mn, and Pb [30]. Saikia et al. [64,89] used LZS instead of 25–35% fine aggregates to prepare mortar. The leaching amounts of Cu, Pb, and Zn exceeded the limits of local waste recycling regulations. Similarly, Barna et al. [134] used LZS to prepare materials for road construction. It was found that when the pH value was greater than 12 or less than 6, a large amount of Pb and Zn would be leached from the LZS. Therefore, when LZS is used to prepare road materials, the leaching of harmful elements can be accelerated when the material meets water, especially in an extremely acidic or alkaline environment [32].

## 6. Conclusions

This paper summarizes the adverse effects of steel slag, copper slag, lead-zinc slag, and electric furnace ferronickel slag in the application of cement-based materials, and the following conclusions can be drawn.

Metallurgical slags can affect the fresh properties of mortar or concrete. The delaying effect of SS as an SCM on setting time is the most obvious. When SS content is set at 30%, the initial setting time and final setting time are prolonged by about 60% and 40%, respectively. CS has the least impact on setting time.When used as SCMs, these metallurgical slags can inhibit the hydration of cement. SS will inhibit the formation of hydration products in cement. Compounds that contain heavy metals in CS or LZS can seal clinker particles, thus limiting the hydration of cement. EFS powder will change the morphology and composition of hydration products in composite paste.The microstructure of hardened paste determines the mechanical properties and durability. Adding some metallurgical slags will degrade the microstructure and eventually decreases the strength and durability. SS as an SCM has the most obvious effect on the porosity, and the porosity increases by nearly 20% when the SS content is 30%. As for mechanical properties, when SS or CS is used as an SCM, the influence on strength is the most obvious. The 28 d compressive strength was reduced by about 20% for both SS and CS at 30% content. For EFS and LZS, they had relatively less effect on mechanical properties.There are some issues that are related to engineering safety when metallurgical slags are used in cement-based materials. The SS aggregates will cause poor soundness. The amorphous silica that exists in EFS will cause volume expansion as a result of the alkali-silica reaction. CS can lead to environmental problems because of the leaching of heavy metals, such as Cu, Pb, Mn, Zn, Ni, Cd, and Cr. At the same time, CS can pose great radiological risks to the surrounding environment. LZS has a leaching risk of harmful elements, such as Pb, Zn, and Cu, especially when used to prepare road materials.

## Figures and Tables

**Figure 1 materials-15-03803-f001:**
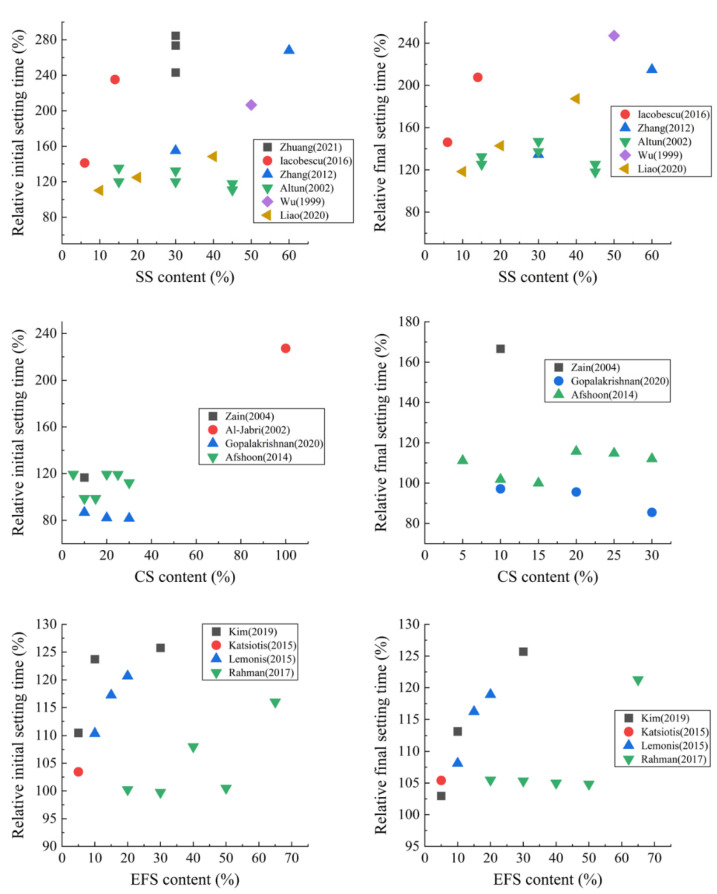
Relative setting time of composite paste containing SS, CS, and EFS. Date from: (1) SS [13,75,79,80,81,82]; (2) CS [76,77,83,84]; (3) EFS [70,71,72,78]. The relative setting time is the ratio of the setting time of composite paste containing metallurgical slag to that of the plain cement paste.

**Figure 2 materials-15-03803-f002:**
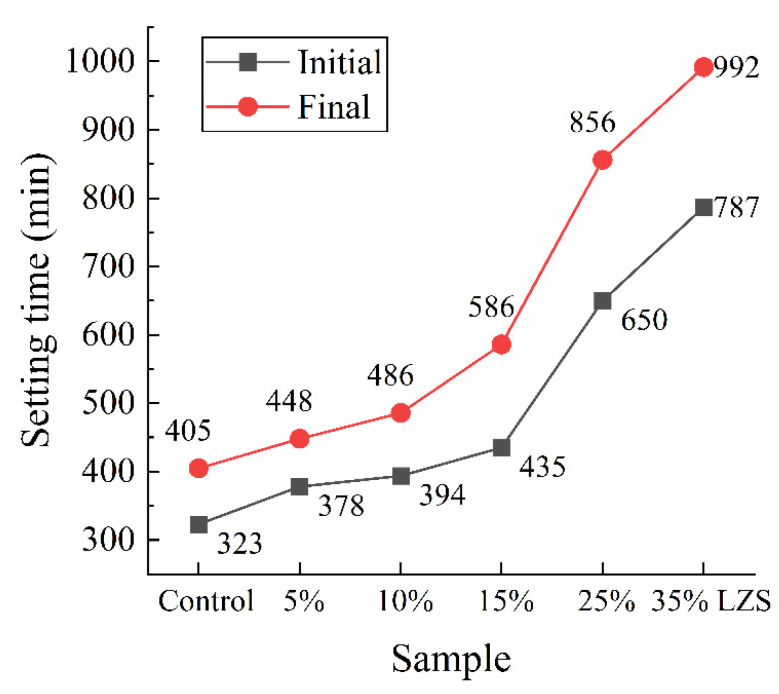
Effect of LZS on the setting time of the mortar [89].

**Figure 3 materials-15-03803-f003:**
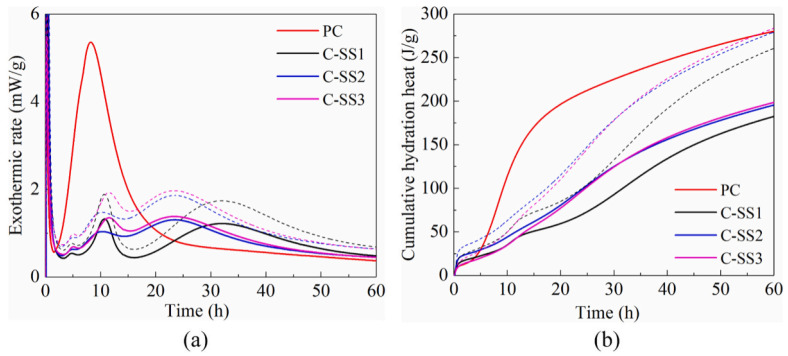
Effect of SS on the hydration heat of composite paste [75]. (**a**) Heat flow; (**b**) cumulative heat.

**Figure 4 materials-15-03803-f004:**
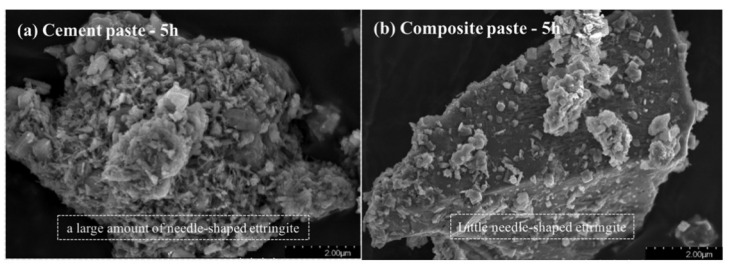
SS inhibits the formation of early-age hydration products, (**a**) the plain cement paste and (**b**) the SS-cement composite paste [75].

**Figure 5 materials-15-03803-f005:**
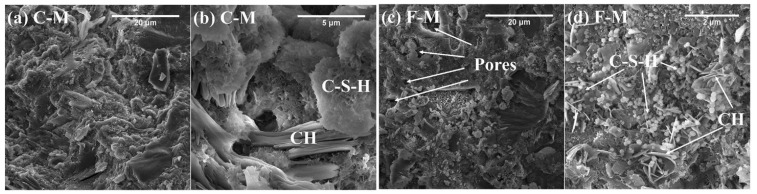
SEM images of the pastes (28 days). (**a**) The cement paste observed in a sacle of 20 μm, (**b**) the enlarged figure of the cement paste (5 μm), (**c**) the composite paste containing 20% EFS (20 μm), and (**d**) the composite paste containing 20% EFS (2 μm) [100].

**Figure 8 materials-15-03803-f008:**
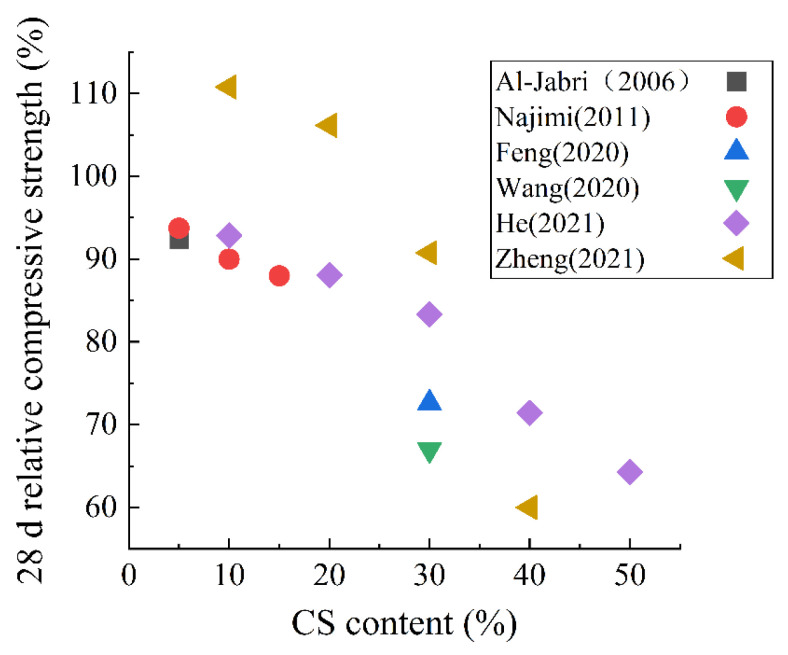
The 28 d relative compressive strength of hardened paste with CS as an SCM [16,27,29,110,111,112].

**Figure 9 materials-15-03803-f009:**
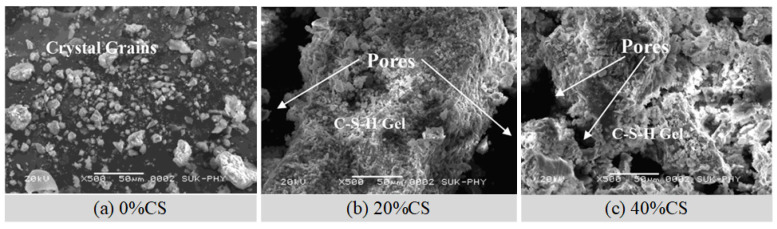
SEM images of the paste containing CS aggregate [94].

**Figure 10 materials-15-03803-f010:**
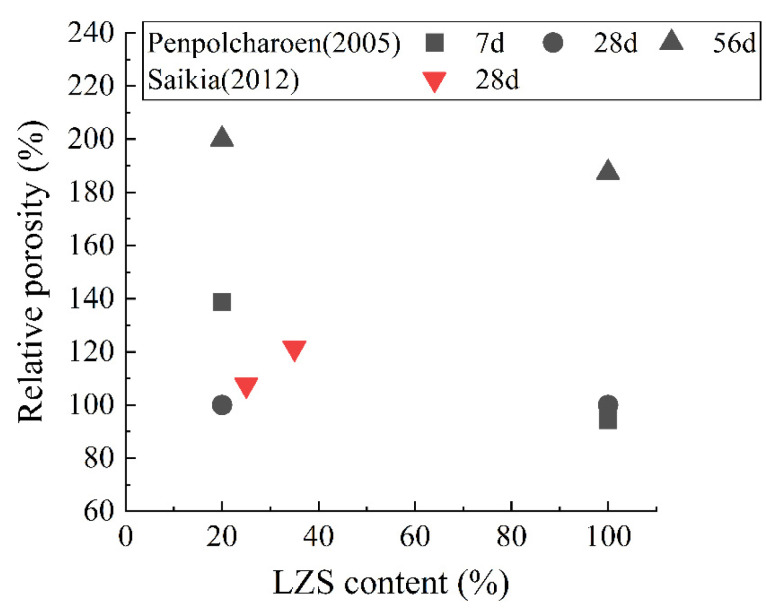
Relative porosity of hardened paste containing LZS aggregate [66,89].

**Figure 11 materials-15-03803-f011:**
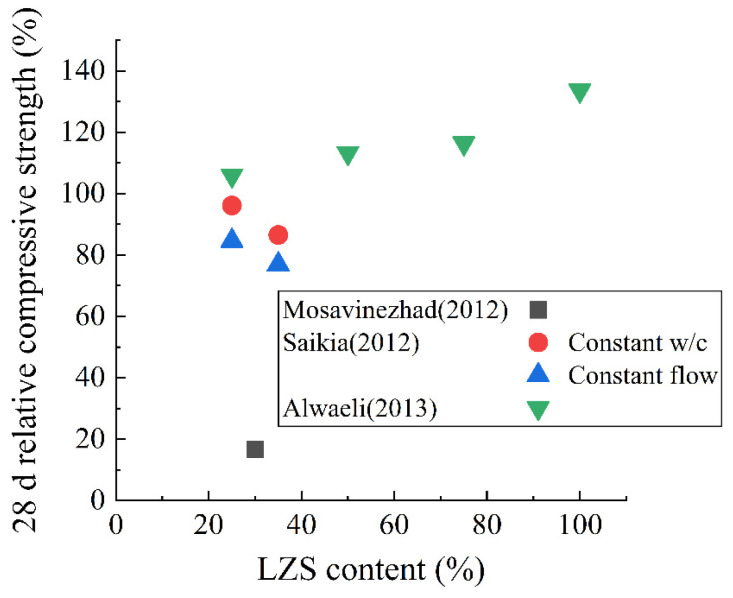
The 28 d relative compressive strength of hardened paste containing LZS aggregate [46,88,89].

**Figure 12 materials-15-03803-f012:**
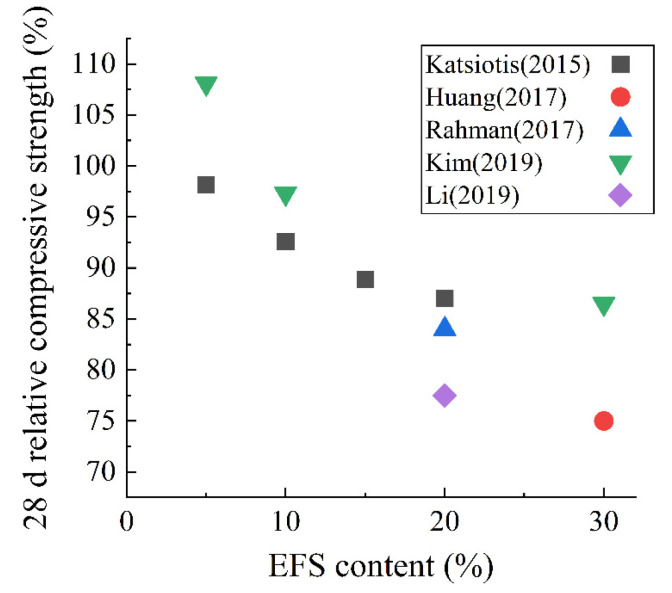
The 28 d relative compressive strength of hardened paste with EFS as an SCM [70,72,78,100,101].

**Figure 13 materials-15-03803-f013:**
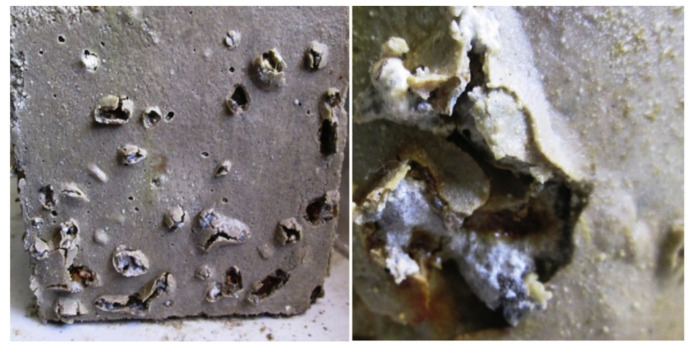
Expansion of hardened paste caused by SS aggregate [91].

**Figure 14 materials-15-03803-f014:**
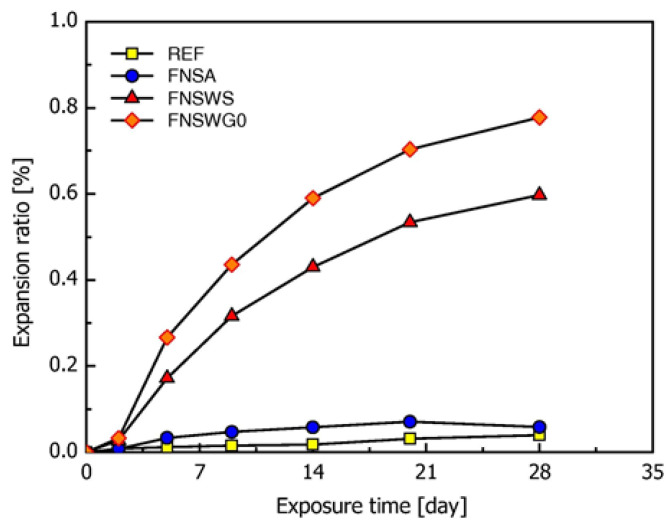
The expansion ratio of mortar containing EFS [127].

**Figure 15 materials-15-03803-f015:**
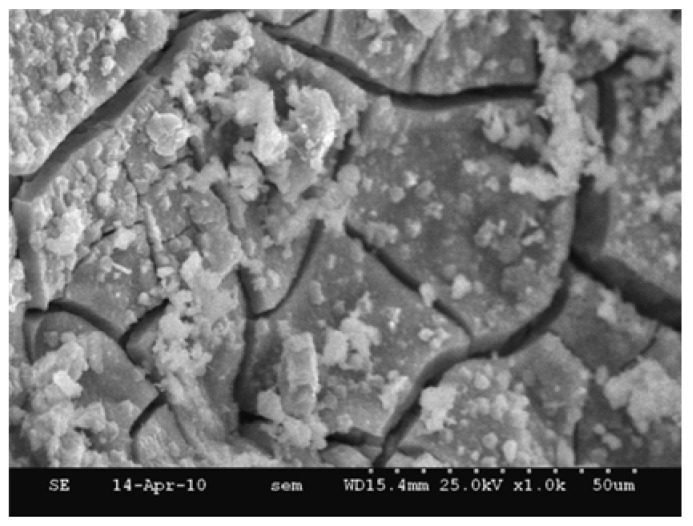
Cracks caused by alkali-silica reaction of paste containing EFS [127].

**Figure 16 materials-15-03803-f016:**
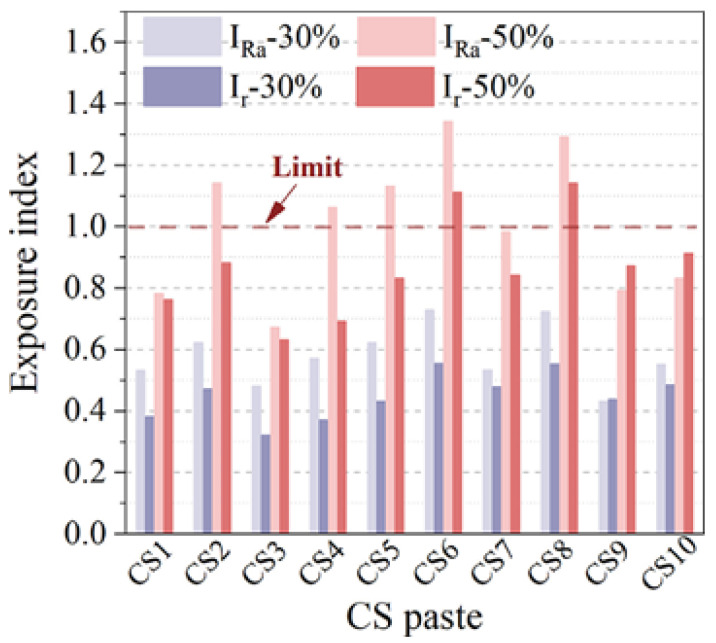
Radioactivity of the mortar containing CS (IRa: internal exposure index; Ir: external ex−posure index) [27].

**Figure 17 materials-15-03803-f017:**
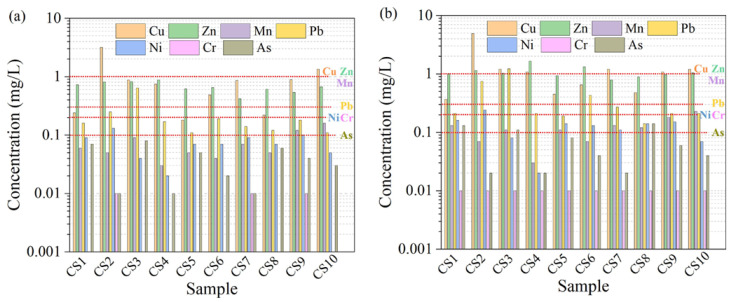
The concentrations of heavy metals leaching from the mortars mixed with (**a**) 30 wt.% and (**b**) 50 wt.% of CS. The red dashed lines are regulated leaching limits [27].

**Figure 18 materials-15-03803-f018:**
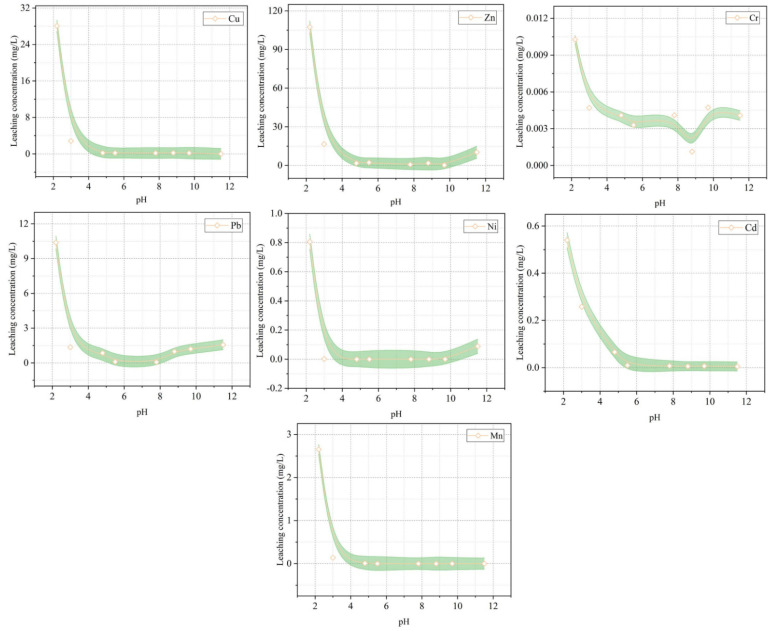
Heavy metals leaching from CS at different pH values [112].

## Data Availability

The study did not report any data.

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
