# Peer review of "Adverse Effects of Using Metallurgical Slags as Supplementary Cementitious Materials and Aggregate: A Review"

_materials, 2022, doi:10.3390/ma15113803_

Round 1
Reviewer 1 Report
This review reports properties of various slags for cementitious materials and aggregate. The readers may not understand some properties in this review. Revise some contents and after moderate revision this review is worthy of publication in Materials as a review.
(1)In Figure 1, the meaning of the dot circles can not be understood from readers.
(2)What is the definitions of relative porosity (%) and 28d relative compressive strength (%) in Figs. 7, 9 and 12. How did the authors determine the perpendicular axis?
(3)How did the authors linearly fit data in Figs. 7, 9, 12, and 13? Did the authors use all data in these figures?
(4)As CS (copper slag) data, why 28d relative compressive strength decreases with increasing CS content? The relative porosity decreases with increasing CS content though.
Author Response
This review reports properties of various slags for cementitious materials and aggregate. The readers may not understand some properties in this review. Revise some contents and after moderate revision this review is worthy of publication in Materials as a review.
- In Figure 1, the meaning of the dot circles can not be understood from readers.
We accept the reviewer’s suggestion, and have removed the dot circles.
- What is the definitions of relative porosity (%) and 28d relative compressive strength (%) in Figs. 7, 9 and 12. How did the authors determine the perpendicular axis?
The relative porosity is the ratio of the porosity of the composite paste containing metallurgical slag to that of the plain cement paste. When the relative porosity is greater than 100%, it indicates that the metallurgical slag deteriorates the pore structure. Similarly, relative compressive strength is the ratio of the strength of the composite paste to that of the plain cement paste.
We have explained these definitions in the manuscript. The revisions are as follow:
“Figure 1. Relative setting time of composite paste containing SS, CS, and EFS. Date from: (1) SS [13,75,79–82]; (2) CS [76,77,83,84]; (3) EFS [70–72,78]. The relative setting time is the ratio of the setting time of composite paste containing metallurgical slag to that of the plain cement paste.” (page 4, line 118-120)
“Figure 6. Relative porosity of hardened paste with SS as an SCM [12,13,102,104]. The relative porosity is the ratio of the porosity of composite paste containing metallurgical slag to that of the plain cement paste.” (page 9, line 223-225)
“Figure 7. 28 d relative compressive strength of hardened paste with SS as an SCM [12,54,80,82,102,103,107]. The relative compressive strength is the ratio of the compressive strength of composite paste containing metallurgical slag to that of the plain cement paste.” (page 9, line 227-229)
- How did the authors linearly fit data in Figs. 7, 9, 12, and 13? Did the authors use all data in these figures?
Yes, we did a linear fit with all the dates. However, we consider carefully that this linear fit is lacking in rigor. We have removed this linear fit to avoid misleading the readers.
- As CS (copper slag) data, why 28d relative compressive strength decreases with increasing CS content? The relative porosity decreases with increasing CS content though.
This is due to different researchers having reached different conclusions. We have removed this figure to avoid misleading the readers, and made changes to the content. The revision is as follow:
“When CS is used as an SCM, different researchers have proposed inconsistent results regarding the effect of CS on the pore structure of the hardened composite paste. The research of Zhang et al. [55] showed that adding CS can increase the porosity of the composite paste, while Gopalakrishnan et al. [77] found that the porosity decreased with the increase in the content of CS.” (page 9, line 231-235)

Reviewer 2 Report
The article is an interesting review about the problems that arise from the utilization of different metallurgical slags. In this line, references are updated being most of them from the last 5 years. However, several questions arise from the reading of the document:
- The major concern is that authors summarize all steel process slags in a single term when there are different types of slags (blast furnace slag, basic oxygen furnace slag, electric arc furnace slags, etc.) with different characteristics that make them usable or not in cements.
- Lines 29-30. What types of materials have shortage?
- Line 31: What technique do you refer?
- Line 38: Different types of steel slags, please clarify. You can find further information about the problems of BOF slags for cement manufacture in https://doi.org/10.1016/j.solener.2019.01.055, reference that can be even cited in your paper.
- Line 39: please provide the quantity generated worldwide.
- Line 54: You can find information about the reutilization of copper slags in the reference https://doi.org/10.3390/met11071032, reference that can be even cited in your paper.
- Line 59: Worldwide?
- General comment, when you refer to slags, it sounds better "generated" than "emitted"
- Line 68: 30 Mt of slag?
- Line 76: what other techniques? accounting for?
- Line 109: why not discussed here?
- Table 2: a row with specific gravity of natural aggregates could be useful for the reader as reference.
- Line 136-138: It is not clear what is reduced, please comment.
- Line 201: is there incompatibility metallurgical slag-cement? Please clarify
- Line 276: MPa instead of Mpa
- Line 311: Not all SS have high CaO free, it is mainly the BOF slag
- Line 337: What kind of radiological effects?
Author Response
The article is an interesting review about the problems that arise from the utilization of different metallurgical slags. In this line, references are updated being most of them from the last 5 years. However, several questions arise from the reading of the document:
- The major concern is that authors summarize all steel process slags in a single term when there are different types of slags (blast furnace slag, basic oxygen furnace slag, electric arc furnace slags, etc.) with different characteristics that make them usable or not in cements.
The reviewer’s suggestion is quite reasonable. This paper discusses the basic oxygen furnace steel slag. We have modified the definition in the manuscript, and recalculated the chemical compositions of the steel slag in Table 1. The revision is as follow:
“The basic oxygen furnace steel slag (abbreviated as SS) is a solid by-product re-leased during steelmaking that accounts for 15-20% of crude steel production [5–8]. The following content is focused on the basic oxygen furnace steel slag.” (page 1, line 37-39)
- Lines 29-30. What types of materials have shortage?
The reviewers’ questions are quite insightful. We have modified the manuscript. The revision is as follow:
“The production of cement is an energy-intensive process and produces large amounts of CO2 emission, which causes serious environmental problems of global warming and climate change [2,3].” (page 1, line 27-29)
- Line 31: What technique do you refer?
The technique refers to the preparation of construction materials from metallurgical slags. We carefully considered that it seemed redundant here. We have revised the content. The revision is as follow:
“Therefore, cement-based products must be made more environmentally friendly [4].” (page 1, line 29-30)
- Line 38: Different types of steel slags, please clarify. You can find further information about the problems of BOF slags for cement manufacture in https://doi.org/10.1016/j.solener.2019.01.055, reference that can be even cited in your paper.
We accept the reviewer’s suggestion, and have cited the recommended references. The revision is as follow:
“The basic oxygen furnace steel slag (abbreviated as SS) is a solid by-product re-leased during steelmaking that accounts for 15-20% of crude steel production [5–8]. The following content is focused on the basic oxygen furnace steel slag.” (page 1, line 37-39)
- Line 39: please provide the quantity generated worldwide.
We accept the reviewer’s suggestion. The revision is as follow:
“Approximately 200 million tons of SS are produced worldwide each year, and China accounts for half of them [9,10].” (page 1, line 39-40)
- Line 54: You can find information about the reutilization of copper slags in the reference https://doi.org/10.3390/met11071032, reference that can be even cited in your paper.
We accept the reviewer’s suggestion, and have cited the recommended references.
- Line 59: Worldwide?
Yes. We have supplemented this information in the text. The revision is as follow:
“According to the statistics, the annual emissions of LZS have exceeded 5.5 million tons in worldwide [30,31], which requires a significant amount of land to stockpile [32].” (page 2, line 58-60)
- General comment, when you refer to slags, it sounds better "generated" than "emitted"
We accept the reviewer’s suggestion, and have revised it. The revisions are as follow:
“For every ton of lead and zinc produced, 710 kg and 960 kg of LZS are generated, respectively.” (page 2, line 57-58)
“The production of 1 ton ferronickel alloy will generate 14 tons of slag [35], and the annual emissions of ferronickel slag have exceeded 30 million tons in China, accounting for 20% of the global production [36].” (page 2, line 67-69)
- Line 68: 30 Mt of slag?
Yes. We can find information about the production of ferronickel slag in the reference https://doi.org/10.1016/j.resconrec.2018.08.002
- Line 76: what other techniques? accounting for?
The ferronickel industry uses two basic smelting technologies: electric furnaces and blast furnaces. However, there is a scarcity of literature to provide specific data on the market share of these two technologies. The electric furnace technique is utilized globally and is now the dominant method in ferronickel alloy manufacturing, while the blast furnace method was formerly employed but is now limited to regions of eastern China due to the lack of nickel-rich minerals and the high demand for ferronickel alloys.
- Line 109: why not discussed here?
Because there has been little research on the setting time of the composite paste containing lead-zinc slag (substitute cement). We have removed this sentence in consideration of the coherence of the paper. The effect of lead-zinc slag aggregate (substitute sand) on the setting time of cement mortar is discussed in section 2.2.
- Table 2: a row with specific gravity of natural aggregates could be useful for the reader as reference.
The fourth column in Table 2 already shows the information on the specific gravity of the natural aggregates.
- Line 136-138: It is not clear what is reduced, please comment.
We accept the reviewer’s suggestion. The revision is as follow:
“The results showed that the slump of fresh concrete was reduced from 135 mm to 75 mm and 20 mm, respectively, reduced by 44% and 85%, indicating that the fluidity of concrete significantly decreased.” (page 5, line 136-138)
- Line 201: is there incompatibility metallurgical slag-cement? Please clarify
We can find information about the incompatibility of lead-zinc slag with cement in the reference https://doi.org/10.1007/s12205-012-1240-2
However, considering the small amount of literature, we have revised the text. The revisions are as follow:
“When metallurgical slags are used as aggregates, the uniformity and compactness of hardened paste are degraded due to the bleeding effect.” (page 8, line 200-201)
“Patil et al. [94] discovered that adding CS caused greater porosity in the microstructure (Figure 9), which is owing to the poor connectivity of CS and cement.” (page 10, line 255-257)
“Mosavinezhad et al. [88] used zinc slag to replace fine aggregate sand by 30%, and the results showed the compressive strength and flexural strength of hardened paste in 56 days decreased by about 88% and 70%, respectively.” (page 11, line 292-294)
- Line 276: MPa instead of Mpa
We accept the reviewer’s suggestion. The revision is as follow:
“When the content of LZS increased from 30% to 50%, the 28 d compressive strength decreased from 39.7 MPa to 25.8 MPa, reduced by 35%; the flexural strength reduced from 7 MPa to 5.6 MPa, reduced by 20%.” (page 11, line 281-283)
- Line 311: Not all SS have high CaO free, it is mainly the BOF slag
We accept the reviewer’s suggestion. The revision is as follow:
“Many studies have shown that the basic oxygen furnace steel slag has a high con-centration of free CaO and MgO.” (page 13, line 317-318)
- Line 337: What kind of radiological effects?
The radionuclides (226Ra, 232Th and 40K) in the cement-copper slag composite pastes. We have added this information in the text. The revision is as follow:
“CS poses a great radiological risk (226Ra, 232Th, and 40K) when the content reaches 50% (Figure 17).” (page 14-15, line 346-348)

Reviewer 3 Report
The article deals with adverse effects of using metallurgical slags in concrete. Despite a very long list of references, the article itself is rather sketchy. Some of the information on adverse effects is presented one-sidedly without mentioning existing research results in the literature indicating the opposite effects of using the discussed slag types. This may be due to the fact that the list of cited articles is extremely geographically uniform and there are almost no articles by authors from outside Asia, mainly China. If the article was to be published in a local, national journal, such a limitation could be understood, but the article is to be published in an international journal of recognized repute, and in this situation it is a very serious blame on the authors for omitting authors from outside Asia (with few exceptions).
Another objection, although of lesser importance, is the use of the term 'reusing' in the title and in the text. In my opinion this word means using again the material that has already been used for some purpose. However, in the case of the slags described here, we are dealing with their initial use. It is true that one can find in the literature studies on reusing e.g. copper slag which was previously used as an abrasive in the surface blast-cleaning process (e.g. DOI: 10.21307/acee-2017-038), but the article does not contain such results. Therefore, I request that the word "reusing" be replaced by "using" in the title and the rest of the text.
I will devote the later part of my review to commenting on the results of the copper slag study, as this is material I am familiar with from research I have participated in. I believe that the authors have not been objective in writing about its adverse effect. For example, the bleeding phenomenon mentioned in several places in the text, which is attributed, among other things, to the use of copper slag, is due not so much to the use of aggregate heavier than sand, but to the wrong consistency of the material or the amount of mixing water used. In tests on concrete with CS, which I had the opportunity to carry out, this phenomenon did not occur even when all the fine aggregate was CS. I also completely disagree with the statement that the addition of CS worsens the strength or durability of concrete, and the following publications prove that the opposite results were obtained (DOI numbers): 10.3311/PPci.14512; 10.21307/acee-2017-038; 10.1016/j.conbuildmat.2010 .06.090; 10.1016/j.conbuildmat.2008.12.013; 10.1016/j.conbuildmat.2014.12.092; 10.1016/j.matdes.2009.12.037. The results they contain show improvements in mechanical properties and properties that determine durability, including air permeability, resistance to chloride ingress, sorptivity, etc. The objectivity of the scientist requires that results contrary to the accepted theses are also taken into account. In some places the authors have fulfilled this condition, but unfortunately in other places this diligence has not been met. This should be corrected. Similarly, the information on radiological risks resulting from the use of CS should be confronted with the results of research contained in DOI: 10.3390/buildings10010001, which indicate that while CS itself shows increased radioactivity, concrete using it meets the requirements formulated for building materials intended for the construction of buildings intended for permanent human habitation.
As far as steel slag is concerned, however, it was with some amazement that I read certain statements which apparently stem from little knowledge of the subject. Ground granulated blastfurnace slag has been used for decades as a replacement for the clinker part of cement and concrete made from such cement is unrivalled by ordinary Portland cement in many applications. Yes, cement with such additives achieves lower early strength and 28-day strength, but the strength over a longer period of time (several months) increases to a greater extent and often achieves higher values than cement based on Portland clinker alone. This is elementary knowledge and I am surprised that the authors try to depreciate SS as SCM.
The paper also contains errors of lesser calibre. For example, in two places the authors mention the harmful effect of Pb contained in CS, while in Table 1 the content of this element equals zero. This inconsistency should be solved somehow.
To sum up, I think that, despite serious defects, the authors should be given a chance to make corrections and, after taking into account the above remarks, consider publishing the article.
Author Response
- The article deals with adverse effects of using metallurgical slags in concrete. Despite a very long list of references, the article itself is rather sketchy. Some of the information on adverse effects is presented one-sidedly without mentioning existing research results in the literature indicating the opposite effects of using the discussed slag types. This may be due to the fact that the list of cited articles is extremely geographically uniform and there are almost no articles by authors from outside Asia, mainly China. If the article was to be published in a local, national journal, such a limitation could be understood, but the article is to be published in an international journal of recognized repute, and in this situation it is a very serious blame on the authors for omitting authors from outside Asia (with few exceptions).
The reviewer’s suggestion is quite reasonable. We have supplemented some relevant references to clarify the contribution of our work. It is not our intention to ignore the studies of the authors from outside Asia. It is a fact that China is one of the countries with the highest emissions of metallurgical slag. As a result, many studies are from China, and they can be used as references.
- Another objection, although of lesser importance, is the use of the term 'reusing' in the title and in the text. In my opinion this word means using again the material that has already been used for some purpose. However, in the case of the slags described here, we are dealing with their initial use. It is true that one can find in the literature studies on reusing e.g. copper slag which was previously used as an abrasive in the surface blast-cleaning process (e.g. DOI: 10.21307/acee-2017-038), but the article does not contain such results. Therefore, I request that the word "reusing" be replaced by "using" in the title and the rest of the text.
We accept the reviewer’s suggestion. The revision is as follow:
“Adverse effects of using metallurgical slags as supplementary cementitious materials and aggregate: A review” (page 1, line 2-3)
“However, they have some adverse effects that could significantly limit their applications when used in cement-based materials.” (page 1, line 12-13)
“SS has a long history of being used as a construction material due to its cementitious properties [15].” (page 1, line 44-45)
“Currently, CS can be used as fine aggregate [21–26] or supplementary cementitious material [27–29]” (page 2, line 54-55)
“LZS can be used as fine aggregate in road foundations [33].” (page 2, line 64)
“However, the using of these metallurgical slags in cement-based materials has several adverse effects that significantly restrict their utilization.” (page 2, line 81-83)
“Figure 1 shows the influences of these metallurgical slags on setting time of composite paste when they are used as SCMs.” (page 3, line 110-111)
“These slags may cause bleeding or segregation of the paste when they are used as aggregates.” (page 4, line 123-124)
“It has been found that EFS causes serious bleeding and segregation when it is used as an aggregate [86].” (page 5, line 128-130)
“It can be seen that the porosity tends to increase when LZS is used as an aggregate. According to the findings of Saikia et al. [89], the porosity of 28 d hardened paste rises from 17.32% to 21.03%, and the water absorption rises from 8.29% to 8.75% when the LZS admixture increases from 0 to 35%.” (page 11, line 286-288)
“When LZS is used in cement-based materials, there is a leaching risk of harmful elements such as Zn, Mn, and Pb [30].” (page 17, line 377-378)
“There are some issues that are related to engineering safety when metallurgical slags are used in cement-based materials.” (page 17, line 407-408)
- I will devote the later part of my review to commenting on the results of the copper slag study, as this is material I am familiar with from research I have participated in. I believe that the authors have not been objective in writing about its adverse effect. For example, the bleeding phenomenon mentioned in several places in the text, which is attributed, among other things, to the use of copper slag, is due not so much to the use of aggregate heavier than sand, but to the wrong consistency of the material or the amount of mixing water used. In tests on concrete with CS, which I had the opportunity to carry out, this phenomenon did not occur even when all the fine aggregate was CS. I also completely disagree with the statement that the addition of CS worsens the strength or durability of concrete, and the following publications prove that the opposite results were obtained (DOI numbers): 10.3311/PPci.14512; 10.21307/acee-2017-038; 10.1016/j.conbuildmat.2010 .06.090; 10.1016/j.conbuildmat.2008.12.013; 10.1016/j.conbuildmat.2014.12.092; 10.1016/j.matdes.2009.12.037. The results they contain show improvements in mechanical properties and properties that determine durability, including air permeability, resistance to chloride ingress, sorptivity, etc. The objectivity of the scientist requires that results contrary to the accepted theses are also taken into account. In some places the authors have fulfilled this condition, but unfortunately in other places this diligence has not been met. This should be corrected. Similarly, the information on radiological risks resulting from the use of CS should be confronted with the results of research contained in DOI: 10.3390/buildings10010001, which indicate that while CS itself shows increased radioactivity, concrete using it meets the requirements formulated for building materials intended for the construction of buildings intended for permanent human habitation.
The reviewer’s suggestion is quite reasonable. We have cited the recommended references and some other relevant studies to make our article more objective. The revision is as follow:
“Because the specific gravity of metallurgical slags is 10-40% higher than that of natural aggregates, metallurgical slags tend to promote segregation when utilized as aggregates. Furthermore, some metallurgical slags deteriorate the microstructure of hardened pastes, resulting in higher porosity, lower mechanical properties, and decreased durability.” (page 1, line 15-19)
“Wang proposed that the replacement rate of CS to sand should be less than 40% to avoid bleeding [85]. This may be attribute to the wrong mixing water content or the lower water absorption of CS (it increases the free water content in the paste) [27].” (page 5, line 126-128)
“When CS is used as an SCM, different researchers have proposed inconsistent results regarding the effect of CS on the pore structure of the hardened composite paste. The research of Zhang et al. [55] showed that adding CS can increase the porosity of the composite paste, while Gopalakrishnan et al. [77] found that the porosity decreased with the increase in the content of CS.” (page 9, line 231-235)
“When CS is used as an aggregate, many studies have proposed a positive effect of CS on the concrete, while some other studies have found an adverse effect. This is related to the different nature of CS from different areas as well as the different mix proportions in different studies. Although copper slag has numerous positive ad-vantages when utilized as an aggregate, its potential negative impacts must not be overlooked. Maharishi et al. [113] found that the adverse effect of CS on concrete was particularly significant when the content of CS exceeded 40%.” (page 10, line 249-255)
“On the aspect of mechanical properties, Al-Jabri et al. [114] and Kubissa et al. [115,116] found that adding CS can effectively improve the mechanical strength of concrete. However, some studies found the opposite results.” (page 10, line 257-259)
“For durability, Kubissa et al. [115] found that CS reduced the sorptivity and improved the permeability resistance of concrete. In contrast, Patil et al. [94] found that the im-permeability, chloride ion corrosion resistance, and sulfate attack resistance of hardened paste decreased after adding CS.” (page 10, line 265-268)
“The environmental problems of copper slag used in cement-based materials are mainly focused on the leaching of heavy metals (Pd, Cd, As, Cu, Zn, etc.) and the potential radiological hazards [128]. Although Kubissa et al. [129] have revealed that the radioactivity of concrete containing CS is low, the potential hazards cannot be ignored.” (page 14, line 342-345)
“Metallurgical slags can affect the fresh properties of mortar or concrete. The de-laying effect of SS as an SCM on setting time is the most obvious. When SS content is set at 30%, the initial setting time and final setting time are prolonged by about 60% and 40%, respectively. CS has the least impact on setting time.” (page 17, line 390-393)
- As far as steel slag is concerned, however, it was with some amazement that I read certain statements which apparently stem from little knowledge of the subject. Ground granulated blast furnace slag has been used for decades as a replacement for the clinker part of cement and concrete made from such cement is unrivalled by ordinary Portland cement in many applications. Yes, cement with such additives achieves lower early strength and 28-day strength, but the strength over a longer period of time (several months) increases to a greater extent and often achieves higher values than cement based on Portland clinker alone. This is elementary knowledge and I am surprised that the authors try to depreciate SS as SCM.
This paper discusses the basic oxygen furnace steel slag, which is generated from the steel refinery process. The ground granulated blast furnace slag mentioned by the reviewer is a byproduct of the blast furnace ironmaking process. Compared with the widespread use of ground granulated blast furnace slag, the use of steel slag is much less.
- The paper also contains errors of lesser calibre. For example, in two places the authors mention the harmful effect of Pb contained in CS, while in Table 1 the content of this element equals zero. This inconsistency should be solved somehow.
We accept the reviewer’s suggestion. We have modified Table 1 by replacing “0” with “-”.
- To sum up, I think that, despite serious defects, the authors should be given a chance to make corrections and, after taking into account the above remarks, consider publishing the article.
Thanks for providing us with this chance to modify. The reviewer’s suggestions are reasonable and objective. We have revised the article according to the suggestions above and added relevant literature.

Round 2
Reviewer 2 Report
The authors have adequately addressed all the proposed comments. This has made the manuscript more comprehensive, so it is now acceptable for publication.
Reviewer 3 Report
Let it go.